# LEARNING GENE REGULATORY NETWORKS UNDER FEW ROOT CAUSES ASSUMPTION

**Panagiotis Misiakos, Chris Wendler and Markus Püschel**
Department of Computer Science, ETH Zurich
`{pmisiakos, wendlerc, markusp}@ethz.ch`

## ABSTRACT

We present a novel directed acyclic graph (DAG) learning method for data generated by a linear structural equation model (SEM) and apply it to learn from gene expression data. In prior work, linear SEMs can be viewed as a linear transformation of a dense input vector of random valued root causes (as we define). In our novel setting we further impose the assumption that the output data are generated via a sparse input vector, or equivalently few root causes. Interestingly, this assumption can be viewed as a form of Fourier sparsity based on a recently proposed theory of causal Fourier analysis. Our setting is identifiable and the true DAG is the global minimizer of the $L^0$-norm of the vector of root causes. Application to the CausalBench Challenge shows superior performance over the provided baselines.

## 1 INTRODUCTION

In this work we provide a novel DAG Learning method for the CausalBench challenge (Chevalley et al., 2022), where the task is to learn a gene regulatory network from gene expression data. We assume that the data are generated from a linear SEM, but we change the data generating process compared to prior work in linear SEMs (Bello et al., 2022; Ng et al., 2020; Zheng et al., 2018; Shimizu et al., 2006). In prior work, linear SEMs can be viewed as linearly transforming an i.i.d. random, dense vector of root causes (as we will call them) associated with the DAG nodes as input and the actual data on the DAG nodes as output. In Seifert et al. (2022a;b) the root causes are considered as a form of spectrum of the DAG data, with the SEM playing the role of the inverse Fourier transform. In our work we consider the spectrum to be (approximately) sparse, i.e., assume few root causes and introduce measurement noise in the output. Intuitively, this captures the situation that the DAG data is produced by few data-generating events whose effect percolates through the DAG.

**Contributions.** Towards this competition we provide the following contributions.

- We provide a closed form solution of the linear SEM equation which expresses the data as output of a linear transform obtained by a reflexive-transitive closure of the DAG's adjacency matrix. In this form, prior work on linear SEMs assumed a dense, random valued input vector of root causes, as we call them.

- We pose the new assumption of the input vector being sparse, i.e. the data are generated from a few root causes.

- We propose a novel algorithm for learning a DAG from data with few root causes. It is called SparseRC and based on the minimization of the $L^1$-norm of the approximated root causes. We provide theoretical guarantees for our method.

- We evaluate SparseRC on the CausalBench competition dataset and show that it offers significant improvement over provided baselines.

The proofs of our theoretical claims, experimental results on synthetically generated data with few root causes and a complete exhibition of our approach can be found in Misiakos et al. (2023).

## 2 MOTIVATION

**DAG.** Consider a DAG $\mathcal{G} = (V, E)$ with $|V| = d$ vertices, $E$ the set of directed edges, and no self-loops. The vertices are sorted topologically and we set accordingly $V = \{1, 2, ..., d\}$. Further, we assume a weighted adjacency matrix $\mathbf{A} = (a_{ij})_{i,j \in V}$ of the graph, where $a_{ij} = 0$ if there is no edge. $\mathbf{A}$ is upper triangular with zeros on the diagonal and thus $\mathbf{A}^d = \mathbf{0}$.

**Linear SEM.** Linear SEMs (Peters et al., 2017) formulate a linear data-generating process for DAGs $\mathcal{G}$. A data matrix $\mathbf{X} \in \mathbf{R}^{n \times d}$ consisting of $n$ data vectors (as rows) indexed by the DAG $\mathcal{G}$ satisfies a linear SEM (Ng et al., 2020; Zheng et al., 2018), with independent random noise samples $\mathbf{N}$, if

$$\mathbf{X} = \mathbf{X}\mathbf{A} + \mathbf{N}. \tag{1}$$

**Transitive closure.** Eq. (1) can be viewed as a recurrence for computing the data values $\mathbf{X}$ from $\mathbf{N}$. Here, we interpret linear SEMs differently by formulating the closed form of ths recurrence. To this end, we define $\overline{\mathbf{A}} = \mathbf{A} + \mathbf{A}^2 + ... + \mathbf{A}^{d-1}$, which is the Floyd-Warshall (FW) transitive closure of $\mathbf{A}$ over the ring $(\mathbb{R}, +, \cdot)$ (Lehmann, 1977), and $\mathbf{I} + \overline{\mathbf{A}}$ the associated reflexive-transitive closure of $\mathbf{A}$. Since $\mathbf{A}^d = \mathbf{0}$ we have $(\mathbf{I} - \mathbf{A})(\mathbf{I} + \overline{\mathbf{A}}) = \mathbf{I}$ and thus can isolate $\mathbf{X}$ in (1):

**Theorem 2.1.** *The linear SEM* (1) *computes data* $\mathbf{X}$ *as*

$$\mathbf{X} = \mathbf{N}\left(\mathbf{I} + \overline{\mathbf{A}}\right). \tag{2}$$

*In words, the data values in* $\mathbf{X}$ *are computed as the output of a linear transform, obtained by the reflexive-transitive closure of* $\mathbf{A}$*, with the noise values* $\mathbf{N}$ *as input.*

This linear transform was considered a causal inverse Fourier transform Seifert et al. (2022a;b), which makes the rows of $\mathbf{N}$ the spectra of the data rows in $\mathbf{X}$. Since $\mathbf{X}$ is uniquely determined by $\mathbf{N}$, we call the latter the *root causes* of $\mathbf{X}$.

**Few root causes.** The equivalence of (1) and (2) motivates us to consider a data generation process that differs in two ways from the prior (2). First, we assume that only a few nodes produce relevant input that we call $\mathbf{C}$, up to low magnitude noise $\mathbf{N}_c$. Second, we assume that the measurement of $\mathbf{X}$ is subject to noise $\mathbf{N}_x$. The equation of generating data $\mathbf{X} \in \mathbb{R}^{n \times d}$ becomes

$$\mathbf{X} = (\mathbf{C} + \mathbf{N}_f)\left(\mathbf{I} + \overline{\mathbf{A}}\right) + \mathbf{N}_s \Leftrightarrow \mathbf{X} = \mathbf{X}\mathbf{A} + \mathbf{C} + \mathbf{N}_f + \mathbf{N}_s\left(\mathbf{I} - \mathbf{A}\right). \tag{3}$$

The root causes $\mathbf{C} \in \mathbb{R}^{n \times d}$ represent the the actual information, i.e., the relevant input data at each node, which then propagates through the DAG as determined specified by the SEM to produce the final output data $\mathbf{X}$, whose measurement is subject to noise. Few root causes means that only a few coefficients in $\mathbf{C}$ are non-zero and the noises $\mathbf{N}_f, \mathbf{N}_s$ have negligible magnitude.

**Example.** We assume a river network, which is naturally represented as a DAG (flow occurs only downstream). The nodes $i \in V$ represent cities, and edges are rivers connecting them. We assume that the cities can pollute the rivers. An edge weight $a_{ij} \in [0, 1]$, $(i, j) \in E$, captures what fraction of a pollutant inserted at $i$ reaches the neighbour $j$. The data $\mathbf{X}$ on the DAG, measure the pollution at each node done every a day. The measurement is the accumulated pollution from all upstream nodes. Within the model, the associated root causes $\mathbf{C}$ show the origin of the pollution. Sparsity in $\mathbf{C}$ means that each day only a small number of cities pollute. Negligible pollution from other sources is captured by noise $\mathbf{N}_c$ and $\mathbf{N}_x$ models the noise in the pollution measurements.

## 3 OUR METHOD

We briefly discuss theoretical guarantees (see Misiakos et al. (2023) for all details) and then present our DAG learning method including the handling of interventions.

**Theoretical Guarantees.** Our setting based on the assumption of few root causes is identifiable as follows:

**Theorem 3.1.** *Assume data generated via the extended linear SEM* (3). *We assume that the root causes* $\mathbf{C}$ *are independent random variables taking uniform values from* $[0, 1]$ *with probability* $p$, *and are* $= 0$ *with probability* $1 - p$. *Then* (3) *translates into a linear SEM with non-Gaussian noise and thus* $\mathbf{A}$ *is identifiable due to (Shimizu et al., 2006).*

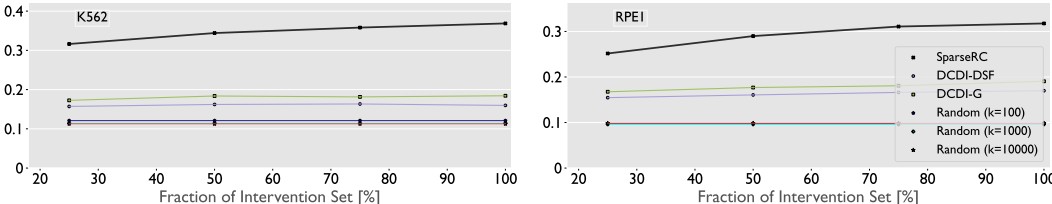

Figure 1: Wasserstein distance metric (higher is better) for the learned DAGs from the datasets K562 (left) and RPE1 (right) (Replogle et al., 2022) based on the CausalBench framework (Chevalley et al., 2022) with varying interventions.

Given the data $\mathbf{X}$ we propose the following optimization problem to retrieve the DAG structure:

$$\min_{\mathbf{A} \in \mathbb{R}^{d \times d}} \|\mathbf{X} - \mathbf{X}\mathbf{A}\|_0 \quad \text{s.t.} \quad \mathbf{A} \text{ is acyclic.} \tag{4}$$

**Theorem 3.2.** *Consider a DAG with weighted adjacency matrix* $\mathbf{A}$. *Given a large enough, but finite, number* $n$ *of samples* $\mathbf{X}$ *the matrix* $\mathbf{A}$ *is, with high probability, the global minimizer of the optimization problem* (4).

**SparseRC.** Our method is formed as the continuous relaxation of the discrete optimization problem (4). We substitute the $L^0$-norm from (4) with its convex approximation (Ramirez et al., 2013), the $L^1$-norm. The acyclicity is then captured with the continuous constraint $h(\mathbf{A}) = tr\left(e^{\mathbf{A} \odot \mathbf{A}}\right) - d$ from (Zheng et al., 2018):

$$\min_{\mathbf{A} \in \mathbb{R}^{d \times d}} \frac{1}{2n} \|\mathbf{X} - \mathbf{X}\mathbf{A}\|_1 \quad \text{s.t.} \quad h(\mathbf{A}) = 0. \tag{5}$$

**Handling interventions.** The gene expression data provided by the CausalBench framework can contain interventions, either for all or for a fraction of genes. An intervention assigns a value to a gene which is independent to the expression data of its predecessors. Mathematically, the linear SEM adopting the intervention scheme is formulated with the following equation ($\odot$ the elementwise product):

$$\mathbf{X} = \mathbf{X}\mathbf{A} \odot \mathbf{M} + \mathbf{N}. \tag{6}$$

The masking matrix $\mathbf{M} \in \mathbb{R}^{n \times d}$ captures the intervention on gene $i$ by removing the incoming edges to node $i$. Thus, $\mathbf{M}$ consists of all ones, except in row $i$, which it is set to zero. In this case gene $i$ is initialized with noise according to (6), or, more generally, with some root cause value together with noise as in (3). Since the positions of the interventions in the dataset are known the optimization problem becomes

$$\min_{\mathbf{A} \in \mathbb{R}^{d \times d}} \frac{1}{2n} \|\mathbf{X} - \mathbf{X}\mathbf{A} \odot \mathbf{M}\|_1 \quad \text{s.t.} \quad h(\mathbf{A}) = 0. \tag{7}$$

## 4 CONTEST EVALUATION

Our method appears to work competitively in synthetic data generated with a few root causes (see Misiakos et al. (2023)) and also in the gene regulatory network dataset by Sachs et al. (2005). In Fig. 1 we present our performance on the gene interaction network benchmark provided by Chevalley et al. (2022). Our method performs better than the provided baselines Brouillard et al. (2020) and also exhibits an upward trend which indicates that it benefits from interventions.

**Implementational details.** For our method, we construct a PyTorch model consisting of a linear layer, which represents the weighted adjacency matrix $\mathbf{A}$. Then, given the data $\mathbf{X}$ processed in batches, and the interventional positions masking matrix $\mathbf{M}$ we train our model with the Adam optimizer with learning rate $\lambda = 10^{-3}$, to minimize the loss in (7). The final adjacency matrix is thresholded at $0.035$ which experimentally showed to result into more than thousands of edges as required by the competition guidelines.

**Conclusion.** We presented a novel form of data generation with linear SEMs based on few root causes and an associated DAG learning algorithms. Our results on CausalBench suggest that the assumption of few root cases may be biologically relevant, which invites further investigation.

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
