# OpenReview forum: "Learning Gene Regulatory Networks under Few Root Causes assumption."
_GSK.ai/2023/CBC_

### Official Review · Reviewer_547T · 2023-04-26

**Rating:** 9
**Confidence:** 5

**Review:**

The authors describe a new data generating framework which generalises the well studied linear SEM model. The framework is based on a causal Fourier analysis which assumes that the data can be impacted by a root causes matrix that is highly sparse. The authors then shows that the adjacency matrix can be recovered with high probability under these assumptions, and then derive an approximation of the optimisation problem. As I understand it, the end optimisation problem is similar to NOTEARS, with the difference that the L1 norm is taken instead of the Frobenius norm. They also include the intervention information through a mask matrix.

The authors then provide an extensive empirical evaluations which validates the proposed framework, both on synthetic data as well as real data. On Causalbench, they obtain highly performing results. More interestingly, this method performs much better than existing linear methods that were tested in the benchmark, which suggests that the assumption of few root causes may be biologically relevant. It would thus be very interesting to further study and hypothesise on how this assumption may correspond to some biological processes.